# Strengths, Weaknesses, Opportunities and Threats (SWOT) Analysis of China’s Prevention and Control Strategy for the COVID-19 Epidemic

**DOI:** 10.3390/ijerph17072235

**Published:** 2020-03-26

**Authors:** Jia Wang, Zhifeng Wang

**Affiliations:** Department of Health Policy and Management, Peking University School of Public Health, Beijing 100191, China; wangjiawst@163.com

**Keywords:** COVID-19, coronavirus, strategy, SWOT analysis

## Abstract

This study used the Strengths (S), Weaknesses (W), Opportunities (O) and Threats (T) (SWOT) analysis method, drawing on our experience of the response to the 2003 SARS epidemic, the 2019 China Health Statistics Yearbook data, and changes in China’s policy environment for the pneumonia epidemic response relating to the novel coronavirus (COVID-19) infection, to perform a systematic analysis of the COVID-19 epidemic prevention and control strategy S, W, O, and T, with a further analysis of a strategic foundation and to determine a significant and relative strategy. We assessed and formulated strength-opportunity (SO), weakness-opportunity (WO), strength-threat (ST), and weakness-threat (WT) strategies for the prevention and control of the COVID-19 epidemic. We conducted an in-depth analysis and identified the highest-priority policies. These are: reshaping the emergency system (SO1); adding health emergency departments to universities and other institutions (WO2); adjusting the economic structure and strengthening international and domestic linkages (ST2); and strengthening public intervention in responding to public health emergencies (WT1).

## 1. Introduction

In December 2019, a new outbreak of pneumonia caused by a novel coronavirus began in Wuhan (Hubei Province, China). It subsequently spread to many countries around the world. The World Health Organization (WHO) announced that COVID-19 is a public health emergency of international concern (PHEIC) on January 30, 2020.

### Background of the Pneumonia Epidemic in China

On December 31, 2019, the Wuhan Municipal Health and Health Committee of Hubei Province, China issued the “Notice of Pneumonia in Wuhan”, after 27 cases of pneumonia had been reported [1]. On January 7, 2020, a China CCTV News Release reported that an expert group had preliminarily identified the “viral pneumonia of unknown cause” as a new type of coronavirus (CoV) [2]. On January 12, 2020, WHO temporarily named the newly discovered CoV “2019-nCoV”. This virus is the seventh identified CoV that can infect humans [3]. On January 25, 2020, the Chinese Communist Party Central Committee designated a group of leaders to manage the response to the epidemic, in order to comprehensively strengthen measures for prevention and control of the virus. On January 31, 2020, WHO announced that 2019-nCoV was a PHEIC [4]. On February 11, 2020, WHO announced that the novel CoV disease was to be named “coronavirus disease-2019 (COVID-19)”, and that the virus that caused the disease was to be named severe acute respiratory syndrome coronavirus 2 (SARS-CoV-2)” [5]. As of 11 March 2020, the COVID-19 epidemic continues to spread. There are more than 80,000 confirmed cases in 34 provinces (regions) of China, and cases have been confirmed in 106 countries, including South Korea, Iran, Japan, Italy, and the United States [6].

This article discusses the rapid spread of the COVID-19 epidemic. Based on our experience of the response to the SARS epidemic in 2003, the 2019 China Health Statistics Yearbook data, and changes in China’s policy environment for the response to COVID-19, herein we present a systematic analysis of the advantages, disadvantages, opportunities, and challenges relating to the current COVID-19 epidemic prevention and control strategy. In addition, we provide suggestions to aid further development of epidemic prevention and control strategies, and scientific decision-making.

## 2. Opportunity Window: SWOT Analysis of COVID-19 Prevention and Control Strategies in China

SWOT analysis refers to the assessment and evaluation of various strengths (S), weaknesses (W), opportunities (O), threats (T), and other factors that influence a specific topic. It comprehensively, systematically, and accurately describes the scenario in which the topic is located. This helps to formulate the corresponding strategies, plans, and countermeasures, which are based on the results of the assessment [7]. This method can be used to identify favorable and unfavorable factors and conditions, solve current problems in a targeted manner, recognize the challenges and obstacles faced, and formulate strategic plans to guide scientific decisions. This study used the SWOT analysis method, and drew on our experience of the response to the 2003 SARS epidemic, the 2019 China Health Statistics Yearbook data, and changes in China’s policy environment for the COVID-19 response, to perform a systematic analysis of the COVID-19 prevention and control strategy.

### 2.1. Strength Analysis

#### 2.1.1. The Medical and Health System is Gradually Improving

According to data from the China Health and Health Statistics Yearbook 2019 [8], the amount of medical resources in China has been steadily increasing. 

The gross domestic product (GDP), the number of medical and health institutions, the number of beds in medical and health institutions, and the number of health technicians per 1000 of the population, have all been increasing year-by-year; therefore, the medical and health system is gradually improving (Figure 1, Figure 2, Figure 3 and Figure 4).

#### 2.1.2. Comprehensive Advancement of China’s Health Emergency System

In 2003, the SARS epidemic had a huge impact on China’s economy and society, especially the lives of ordinary people. At the time of the SARS outbreak, there were no specialized health emergency agencies, established plans, systems, mechanisms, or applicable legal framework. Faced with severe challenges, the Chinese government responded by developing a health emergency system based on a “one plan, three systems” strategy, and continued to strengthen basic health services [9]. “One plan” refers to the emergency plan, and “three systems” refers to establishing and improving (1) emergency response systems, (2) mechanisms, and (3) legal systems. In the face of the rapidly advancing COVID-19 epidemic, the Chinese health emergency system can play a role in the guiding, coordination, prevention, and control of this epidemic (Table 1).

#### 2.1.3. Quick and Effective Cooperation in the Joint Prevention and Control of Various Departments

Multisectoral cooperation is complex. China has a centralized and unified administrative leadership system. In response to major public health emergencies, the administrative execution of each department is robust. In the face of the COVID-19 epidemic, China has launched a joint prevention and control mechanism, which has played an important role in defining each department’s response. The Department of Trade needs to increase the supervision of key trading places, especially places involved in the trade of poultry and wild animals, such as farmers’ markets, bazaars, and supermarkets. The Department of Transport needs to implement traffic control measures in key places such as stations, airports, wharfs, and closed public transport, and to ensure that necessary measures, such as ventilation, disinfection, and temperature monitoring, are observed once a suspected case of COVID-19 is identified. The Human Resources and Social Security departments need to strengthen their handling of labor relations during the epidemic period, protect employees’ wages and benefits, and make arrangements for employees to take paid leave. On this leave, according to a type of subsidy standard, each person will be given a subsidy of 29 or 43 U.S. dollars per day [18]. The Education Department requested that schools establish a current ledger for teachers and students, and has postponed the start of the spring semester, which was maybe scheduled to start in April or May 2020. The national government has allocated 4.4 billion yuan to the COVID-19 prevention and control subsidy fund to support various agencies performing epidemic prevention and control related work; an additional 500 million yuan has been allocated to Hubei Province, and the Finance Department and the Medical Insurance Department have identified further costs associated with the investigation and outpatient medical treatment of the COVID-19 epidemic. Policies to cover the medical expenses of patients diagnosed in other places should also be put in place. In the event of a major illness, health insurance covers some medical expenses in accordance with regulations; the personal burden will be subsidized by the finances. The required funds are paid in advance by the medical provider, and the central government reimburses the medical provider at 60% of the actual expenses incurred [19]. The establishment of a national temporary reserve system for the prevention and control of materials should be promoted, guiding e-commerce platforms to control prices, flow, and reasonable placement. In addition, a supply and demand mechanism for e-commerce is necessary, as are radio and television media departments for the general public to obtain epidemic information reports and enrich their home life; free coverage for users to open cable channels for one month should be provided (Figure 5).

### 2.2. Weakness Analysis

#### 2.2.1. Cases of COVID-19 Developed in Many Regions within a Short Period

In 2019, the first 27 confirmed cases of COVID-19 were officially notified; of those, 7 patients were in serious condition, and the remaining patients were stable [1]. However, COVID-19 spread to 34 provinces or regions of China within one month, and the total number of countries affected worldwide is increasing [5]. A lag in the release of information, especially compared to the SARS epidemic, may be one of the reasons for the rapid spread of COVID-19 (Table 2).

#### 2.2.2. The COVID-19 Epidemic Coincided with the Spring Festival, Complicating Epidemic Prevention and Control Measures

The 2020 Railway Spring Festival Transport started on January 10, 2020 and was scheduled to end on February 18, 2020, after a total of 40 days. The COVID-19 outbreak coincided with this transport period. During Spring Festival use of all modes of travel, including airlines, railway, road, and water increases, providing opportunities for the virus to spread. Of note, the windows of trains, planes, subways, and passenger cars are closed.

#### 2.2.3. China is a Vast Country with a Huge Population

China’s land area is about 9.6 million square kilometers, and there are 34 provincial-level administrative regions, including 23 provinces, five autonomous regions, four municipalities, and two special administrative regions. By the end of 2018, China’s population had reached 1.395 billion [20], and China’s floating population including temporary residents, registered tourists, and transient populations reached 241 million [21]. Huge population bases, large-scale crowd movements, and large regional movement diameters have posed great difficulties to the epidemic prevention and control efforts.

#### 2.2.4. Lack of Relief Materials and Human Resources

The discrepancy between supply and demand of prevention and control materials remains prominent, and there have been serious shortages of masks and protective clothing in Wuhan and other places in Hubei Province. China has used the central reserve to ensure the needs of the epidemic area are met, and has adopted methods such as increasing production capacity and strengthening international cooperation, in order to meet the current mismatch between supply and demand. Through these efforts, the resumption of work and production has reached 40%, but the demand for protective clothing, masks, and other materials remains high. Hubei’s requirements include roughly 100,000 pieces of medical protective clothing per day, and 3 million pieces per month. However, there are only 40 companies that meet China’s standard production capacity licensing standards, spread across 14 provinces. The total production capacity is only 30,000 sets per day, and the imbalance between supply and demand persists [22]. There are also serious shortages in the provision of beds and human resources in medical and health institutions, with varying treatment capabilities. Wuhan Epidemic Prevention Command completed Wuhan Lei Shenshan Hospital February 3, 2020, adding 1,300 new beds to ease the shortage of medical treatment beds. By 11 March 2020, 346 medical teams have been dispatched from all over the country, and more than 42,600 medical personnel have arrived in Hubei Province. Departments include respiratory, infectious, and ICU critical care departments. There are local medical teams, the army, Western medical staff, and Chinese medical staff.

#### 2.2.5. Health Emergency Discipline is Underdeveloped

In China, there is no health emergency discipline. The training and reserves of health emergency personnel need to be greatly strengthened. Universities do not offer training in health emergency response as a medical specialty, and there are few health emergency specialists in the country. Since the start of the COVID-19 epidemic, nearly 800 public health workers have been recruited in several capital cities and large cities in China, leaving a major shortage of health emergency professionals.

#### 2.2.6. The Public is Flustered and Lacks Awareness

Although the epidemic prevention work has attracted the attention of the whole world, there are still some places where individuals do not fully understand the necessity and importance of prevention and control. Some people are “playing boldly” and being defiant in the face of the epidemic, refusing to wear masks, and laughing at those who do. Some people take their children into crowded places and travel long distances without protection. Some departments are not sensitive to the epidemic situation, are afraid of trouble, and do not implement the requirements of occupational exposure protection or fever screening, as well as other prevention and control requirements. Others believe that epidemic prevention is a matter for the health and disease control departments, and their response to the “joint prevention and joint control” movement is slow. These phenomena reveal that there are still loopholes and shortcomings in the public health system, and that health literacy and disease prevention knowledge in the general population requires strengthening.

#### 2.2.7. Rumors

Methods of prevention of COVID-19 has become the topic of paramount concern to the general public. Along with the public’s attention to the epidemic, various rumors have also emerged, e.g., that *Isatis indigotica* (a plant used in Chinese traditional medicine), saunas, and antivirals, such as the flu drugs oseltamivir and ribavirin, can prevent CoV infection; and that drinking high-alcohol liquor, smoking, or fumigation with vinegar and salt water can be used as treatment. None of these rumors have any solid foundation in science or medicine.

### 2.3. Opportunity Analysis

#### 2.3.1. New Exploration of the Pneumonia Epidemic

SARS-CoV-2, has a high level of infectivity, which poses challenges for prevention and control of infection. SARS-CoV-2 is a newly emergent CoV from an independent evolutionary branch [23,24]. It belongs to the same beta coronavirus genus as the original SARS-CoV and MERS-CoV, but it is genetically distinct from these two related viruses, with a nucleic acid homology of <80%. The virus that is currently known to be the most closely related to SARS-CoV-2 was isolated from a Yunnan chrysanthemum bat and shares a nucleic acid homology of 96% [25]. Therefore, the chrysanthemum bat may be the original source of SARS-CoV-2, although the direct source of the current epidemic has not been found. Chinese researchers have been quick to respond to the epidemic by initiating new research on the epidemiology, clinical characteristics, treatment, clinical outcomes, and laboratory and radiological characteristics of COVID-19, carrying out research and development of diagnostic tests and candidate vaccines for SARS-CoV-2. Multiple studies published in the world’s top journals, including a series of articles that were published in the *Lancet* less than 2 months after the start of the epidemic [26,27,28,29] to promote identification of the source of infection, the duration and mechanisms of human transmission, and clinical knowledge about the diseases, show the need for more in-depth research on COVID-19.

#### 2.3.2. Further Improvement and Inspection of the Emergency Health System

The COVID-19 epidemic is a severe test for China’s health emergency system, and challenges the Chinese government, health departments, medical institutions, and disease prevention and control departments. Insufficient long-term reserves of materials, lack of training of health emergency personnel, and the ability to detect and respond to emergency medical treatment all need to be addressed. Further data is required to improve knowledge on the etiology, detection and monitoring of important infectious disease pathogens, virus-host interactions, pathogenic mechanisms, antiviral drugs, and vaccines. Due to constraints such as limited data and knowledge of the pathogen and experimental conditions, there has been insufficient long-term deployment of research, especially in the research of some severe infectious diseases; a lack of integration of resources and sharing of results also exists.

#### 2.3.3. Opportunities for Education in Infectious Diseases

Public awareness of the COVID-19 epidemic is generally high, which provides a good opportunity for national education on infectious diseases. Through official channel information release, the public became aware of the COVID-19 epidemic. For example, Baidu index big data shows that the average daily number of internet searches of the keyword “masks” is 3.9 million. At the same time, related issues such as “the correct use of masks” and “N95 protective masks” have also become a major source of interest among the general public. “Symptoms of novel coronavirus”, “transmission routes of novel coronavirus”, and “new cases of coronavirus” are also internet search terms that have been trending, revealing a high level of concern. Public awareness of epidemic prevention and control has been improved by the COVID-19 epidemic, as has health education overall.

### 2.4. Threat Analysis

#### 2.4.1. Unknown Source of the Pneumonia Outbreak at the Start of the COVID-19 Epidemic

At present, the source of the original pneumonia outbreak at the start of the COVID-19 epidemic, and the transmission mechanism of the virus are unknown, and there is no specific treatment available. At the time of completion of this manuscript in early March, 2020, the COVID-19 epidemic is still rapidly spreading around the world.

#### 2.4.2. Impact of COVID-19 on Public Daily Life, Work, and Psychology

During the national extended spring holiday period, most enterprises and institutions suspended production and operations, reopening of various schools has been delayed, some railways and flights have been suspended, and many people have been isolated at home.

#### 2.4.3. Impact of COVID-19 on the National Economy

From the current policy adjustments of various national departments and changes in public life and work plans, the main domestic industries that affect the country include: service industry (including accommodation, tourism, and catering), infrastructure (including construction machinery, transportation investment, and power heating), and transportation (including rail, plane, and road). Due to international restrictions, the cancellation of international flights may cause some fluctuations in import and export trade.

According to the above SWOT analysis, the relevant factors for COVID-19 epidemic prevention and control strategy in China are shown in Table 3. Based on the SWOT analysis results shown in Table 3, further analysis provides a basis for obtaining a strategy:

a)Improving the national public health emergency management system is a long-term development goal.

China has integrated the modernization of national governance into the specific work of economic, political, cultural, social and ecological civilization construction. The prevention and control of major public health emergencies is an important part of the field of social governance, and has become a major challenge to advance the modernization of the national governance system and governance capabilities. Any major public health emergency threatens all human beings, regardless of country, region, race, or population. This is a common challenge facing humankind.

b)Rejuvenating the country with science and technology requires that talents are cultivated

The cultivation of public health disciplines and talents in China should be encouraged. It is necessary to further train high-level experts in public health epidemiology, strengthen the teaching and research of acute infectious diseases (or emerging infectious diseases) in various public health colleges, and establish master’s, doctoral and Postdoctoral programs. In addition, strengthening interdisciplinary specialties such as health emergency, health management, and health policy in professional settings and changing the current state of marginalization of related specialties and disciplines.

c)China’s economic recovery

Judging from the current development trend of the COVID-19 epidemic, the situation has and will have great impact on China’s economic and social development. Since the epidemic occurred during the traditional Spring Festival, the economic impact affects consumer demand, production and life, industrial chain reconstruction, and investment stability. Consumption during the Spring Festival in 2020 will be greatly reduced. Among them, catering, hotels, tourism, entertainment, department store transportation, education and other traditional living service industries have suffered the most. Some small and medium-sized manufacturing enterprises have difficulty operating, and there is insufficient protection and supply of agricultural production materials for spring cultivation.

d)Chinese public psychology should be strengthened with intervention

The COVID-19 epidemic is a double test of the public body and mind. The first stage is the immediate response stage, that is, the beginning of the epidemic. The general public is still in a state of panic, denial and disbelief; the second stage is the complete response stage, that is, the rapid development of the epidemic. The general public is excited and anxious, and has negative emotions such as pain, confusion, and anger. The third stage is the eradication phase, which aims to eliminate the negative emotional impact and psychological shadow that the epidemic has brought. It can be said that due to the impact of this epidemic, the complete recovery of public psychology will be a “marathon.” Even after the epidemic is over, bad moods may still affect daily life.

## 3. COVID-19 Infection Prevention and Control Strategy for Pneumonia

Based on the SWOT analysis of the COVID-19 epidemic, a strategic opportunity window analysis model was constructed. We integrate the current prevention and control strategy of the COVID-19 epidemic in China instead of fragmentation. We have a more systematic and intuitive understanding. Based on detailed analysis of the above SWOT steps, we performed data mining, and extraction resulted in actionable priority plans to scientifically support the priorities. The highest priority strategies are reflected in SO1, WO2, ST2, and WT1. Therefore, SO1, WO2, ST2, and WT1 are explained in more depth. (Table 4)

### 3.1. Strength-Opportunity (SO) Strategy

#### 3.1.1. Continuous Reshaping of the Health Emergency System

The COVID-19 epidemic exposed the shortcomings of China’s public health emergency management system. The top-level planning and design of the emergency management system for major public health emergencies is outdated, with a centralized system for managing major public health emergencies. This has led to a diminished level of management effectiveness, and a lack of resilience. The public health system and basic security facilities are lagging behind and have not kept up with the level of economic development, including the lack of early public health emergency prevention and control plans; uneven data sharing and transformation application channels; weak wildlife market supervision; a lagging legal framework that is difficult to implement; weak health emergency resources census database; lack of specialized emergency personnel for major public health emergencies; lacking composite rescue teams; insufficient emergency rescue teams; insufficient professional emergency rescue funding; and a serious shortage of human resources. The COVID-19 epidemic in China provides a major opportunity for the reform and development of a public health emergency management system.

The core competency of public health emergency management is the health emergency response system. However, the strategic positioning of the core competency of public health emergency management and the improvement in planning and construction are not clear. In recent years, the Chinese government has proposed developing a health emergency system and improving emergency response capabilities. The process of the progression from the emergence of the pneumonia to the rapid spread of the COVID-19 epidemic in China is a major test of the national governance system and governance capabilities. From the perspective of theoretical research, there is still no formal and authoritative definition of the connotation and extension of the core competence of public health emergency management in China. From the perspective of practical challenges, China’s public health emergency management has revealed much inconsistency, and gaps in awareness, philosophy, governance, command, coordination, and action. The COVID-19 epidemic in China revealed a limited degree of response during the early stages of the outbreak; the performance of the authorities in different regions in the prevention and control of the epidemic was variable across regions, and the use of material donations caused public doubts and social response; Ineffective measures by individual local governments resulted in ineffective control of local epidemics.

In summary, to implement the above ideas in concrete terms, we must answer a series of questions: What major deficiencies have emerged in the public health emergency management system in the face of major public health emergencies, such as the COVID-19 epidemic in China? What is the core mandate of the public health emergency management system? In the process of improving the national public health emergency management system, what foreign experiences can we learn from? What is the core competency of public health emergency management? What are the key processes for improving public health emergency management core capabilities?

The national health emergency system needs continuous development and remodeling. In particular, we recommended exploiting the core capabilities of public health emergency management, clarifying its strategic position and planning, developing and upgrading processes related to the fate of the country and its people, and the current and future systems are in urgent need of long-term strengthening. The establishment of a professional team of university health emergency teachers, discipline construction, and a personnel training system should be included in the scope of the national health emergency system. Restructuring organizations can lead to the establishment of health departments where required. Establish a series of square cabin hospitals with different medical or technical support functions to ensure emergency rescue tasks for public health emergencies by Health administration [30]. Emergency management positions could be used to play a better role in communicating and coordinating the joint prevention and control strategy; the system plan should include prevention and control plans for major infectious diseases during special periods such as the Spring Festival and ban wildlife trade. The quarantine institutions, detection technologies and treatment options all require further improvement. Hazard risk assessments, hazard grading systems and corresponding response mechanisms among different departments for major public health emergencies also require improvements. Market supervision must also be strengthened, and online as well as offline wildlife trading must be banned by State Administration of Market Supervision. The production capacity of domestic health emergency protection materials is insufficient, and the international and domestic health emergency material standards are inconsistent; thus, it is difficult to “benchmark” the quality of these materials. The Ministry of Industry and Information Technology should formulate applicable standards for materials. Transportation of protective materials should also be improved. There is an urgent need to reshape the national health emergency system in the near future and to address these issues in the reshaping.

#### 3.1.2. People-Oriented and Value Policy Orientation

The construction of overall management systems should be “people-oriented” with regards to the level of protection of medical and epidemic prevention personnel. After a large number of medical and health and epidemic relief teams have been put in place, the overall management of food, accommodation and protection needs to be enhanced. One should be more humane and pay attention to policy orientation. For example, more public interests should be considered in the aspects of refunds and reissues of civil aviation and railway tickets. Many individuals have expressed their demands through the internet, and many have already responded to national policies for refunds. The refund fees amount to thousands of yuan.

#### 3.1.3. Integration and Upgrade of the Health Emergency Information System

The health emergency information system needs to be integrated and upgraded, to strengthen the national-level surveillance of major infectious diseases, and to formulate a centralized reporting and information release system for new or unknown infectious diseases. The epidemic situation must be reported and released in a timely and accurate manner to avoid concealment, duplicate reports, omissions, and misreporting. In addition to publishing dynamic information on the epidemic, information platforms should allow the development of statistical analysis functions.

### 3.2. Weakness-Opportunity (WO) Strategy

#### 3.2.1. Formulate a Health Emergency Response System for Major Infectious Diseases during Holidays

When the occurrence of a major infectious disease epidemic coincides with a statutory holiday, preparations should be made to respond to public health emergencies over holidays, and the statutory holiday or series of holidays should be terminated temporarily in advance.

#### 3.2.2. Establish a Health Emergency Department in Universities

There are only a few universities in China, such as Jinan University, Henan Polytechnic University, and Jiangsu University, which have undergraduate emergency courses, and there are very few professionals qualified to lecture or conduct scientific research in the health emergency field. Regardless of improvements in national health emergency awareness, professional training of national health emergency administrative personnel, or from the improvement of national health emergency scientific research level, setting up health emergency disciplines in colleges and universities, and training professional health emergency teachers. It is one of the countermeasures for long-term sustainable development. The COVID-19 epidemic may provide the catalyst needed to take action to develop public health emergency management infrastructure. Not only medical schools, but other universities may also consider setting up health emergency disciplines and forming a team of health emergency teachers. All universities have the educational obligation of cultivating students to learn emergency and establishing emergency awareness. Educational channels for school health emergency can also strengthen publicity for social health emergency.

From the perspective of long-term development, with the special needs of the country as the guide, additional health emergency majors have been established in universities, the number of health emergency professional teachers has been increased, and the development of health emergency disciplines is imminent. Scientific research on health emergencies can be strengthened to reinforce the country’s scientific research output for health emergencies, to support the capacity of health emergencies, and to popularize the basic knowledge of public health emergency prevention and control for students, as well as providing basic guarantees for universities to respond to public health emergencies. In addition to the establishment of independent health emergency departments in health departments, medical and health institutions, disease prevention and control agencies, and health supervision agencies, health emergency management departments should also be established within other joint health emergency prevention and control agencies.

#### 3.2.3. Construction of the Health Emergency Culture and Code of Conduct of the System

It is important for emergency health education to be conducted, as well as the formulation of a “Code of Conduct for Health Emergency Response”. Training exercises should be carried out on community health emergency knowledge, field training of health emergency professional teams, long distance training as well as quality development, and promote health emergency awareness through banner slogans and display boards. Through the cultural construction activities of health emergency communities and departments, such as “health emergency team song collection activities”, “telling health emergency stories” and “remembering emergency training diaries”, the content of health emergency culture is enhanced, health emergency awareness is established, and health emergency personnel are stimulated. Furthermore, a sense of honor and achievement can form an effective spiritual motivation. Attention should be paid to the coordination of urban governance and health emergency policies, and it should not be controlled by public opinion, in order to avoid excessive interference in urban governance.

### 3.3. Strength-Threat (ST) Strategy

#### 3.3.1. Authoritative Departments Releasing Accurate Information in Time

A national unified platform for the release of epidemic information should be established. Authoritative departments should publish accurate information in a timely manner. At the same time, an epidemic information release and review system should be formulated. The provinces should centrally report epidemic information by infectious disease direct reporting system. National authorities should respond to public concerns immediately.

#### 3.3.2. Timely Adjustment to the Economic Structure and Strengthening of International and Domestic Regional Linkages

The COVID-19 epidemic spreads rapidly around the world. The epidemic triggered not only the global public health crisis, but also global public crises that penetrated into many fields such as politics, economics, culture, and health. In responding to COVID-19 epidemic, the shortcomings of public health emergency management systems have become fully exposed in multiple countries. The epidemic has exposed the insufficient knowledge of major public health emergencies, limitations in prevention measures, and limitations in core competencies in public health emergency management. The global public health emergency management system cannot meet the needs of the current complicated and severe epidemic situation.

Iran was the first country in the Middle East to experience an outbreak of COVID-19. The Iranian economy has been subject to economic sanctions for a long time, and the government cannot easily impose a shutdown in the country. If the outbreak in Iran continues to grow, it may cause more serious health incidents in the Middle East as a whole; Japanese domestic enterprises and local governments are in a state of emergency. The Japanese government has taken measures to start budget reserves to help small and medium-sized enterprises to manage their financial shortfall.

To adjust the economic structure in a timely manner, people-intensive production enterprises should strengthen the impact assessment, strengthen communication, and linkage with international organizations and other countries. They should also share information, actively respond to adverse measures of international organizations and other countries and strengthen entry-exit inspection and quarantine. Efforts will be made to control population flow in epidemic-stricken areas, strengthen the sharing and release of epidemic information internationally, and to establish an international cooperation and coordination mechanism for health emergency rescue.

#### 3.3.3. Enhance Scientific Research and Transformation of Major Infectious Diseases

The long-term accumulation of basic research on the pathogens of major infectious diseases will provide the theoretical basis and supporting technology for the detection, diagnosis and control of major infectious diseases. Strengthening of the scientific research capabilities for infectious diseases, laboratory construction, timely conversion of production capacity, and strengthen the linkage and upgrade of upstream and downstream scientific research institutes and enterprises, are all necessary. Research institutions and institutions of higher education should continue to support studies exploring the etiology of important pathogens, pathogenic mechanisms of infection, animal models, antiviral drugs and vaccines, as well as research on detection and monitoring technologies, and control technologies for severe and foreign pathogens. Support is also crucial for research into mechanism and control technology. Research on prevention and control strategies and measures, technical support, and drug reserves for responding to public health emergencies is also vital.

#### 3.3.4. Develop the Health Emergency Function of the Medical and Health System

It is necessary to fully develop the functions of the medical and health system, invest in health emergency reserve funds, optimize the allocation of medical and health resources, set up full-time emergency departments and positions in medical and health institutions, increase basic training for medical personnel on major infectious diseases, and enhance pathogen detection in medical and health institutions. The ability to increase the importance of medical personnel on major infectious diseases is required, as is an improvement in the ability to identify and judge early, and strictly regulate the system of surveillance, early warning, and reporting of major infectious diseases. Investment in health emergency reserve funds to enhance defense capabilities is crucial.

### 3.4. Weakness-Threat (WT) Strategy

#### 3.4.1. Strengthening the Public’s Psychological Intervention in Responding to Public Health Emergencies

The COVID-19 epidemic is identified as a Class B infectious disease in China, and measures for the prevention and control of Class A infectious diseases are adopted. Class A infectious diseases include plague and cholera. Class B infectious diseases include infectious atypical pneumonia, AIDS, viral hepatitis, and polio. Patients will experience varying degrees of stigma during the epidemic, and this will cause anxiety, depression, hostility, and other mental and psychological symptoms requiring timely intervention to avoid the emergence of mental and psychological disorders such as long-term post-traumatic stress disorder. With the implementation of the COVID-19 epidemic prevention, control, and governance measures, problems in public health ethics and psychology have become apparent. Public health ethics is to establish reasonable boundaries and standards for citizens to make the necessary sacrifices. Volunteers entering the community and actively conducting psychological assistance can help the public to prevent and control infectious diseases. It is a key strategy to solve public psychological problems.

Based on the monitoring and information management of major infectious diseases, it is important to establish and improve a social psychological early warning system, to strengthen the public’s psychological health education, to open a psychological counseling hotline, unite health emergency professionals and psychological counselors, and provide psychological counseling and emergency interviews to the public. Other interventions, such as cognitive therapy, behavioral therapy, and other professional psychological interventions, are also of crucial importance.

#### 3.4.2. Formulate Return to Work Plans for Different Industries

According to the characteristics of individual industries, plans for returning to work in different industries should be formulated. Non-life security service industries and consumer entertainment service industries can postpone the return journey; however, economic pillar industries should formulate effective response plans to avoid excessive impact on the wider economy.

#### 3.4.3. Increase Support for Health Emergency Education

Guided by the national health emergency manpower demand, one should consider increasing professional emergency health education support, fully developing the role of health emergency employment guidance, adjusting the professional settings and enrollment plans of universities, creating health emergency unified recruitment and sub-specialties, and increasing health emergency professional employment. A system for targeted employment in health emergencies should also be developed.

#### 3.4.4. Implementation of Mobile Offices during Major Infectious Diseases

At present, most enterprises and institutions have postponed work, and the implementation of mobile offices can reduce economic losses and the spread of major infectious diseases caused by contact with people in the workplace A mobile office plan for major infectious diseases should be formulated, and qualified units should equip staff with internet-enabled smartphones or tablets. Scientific research institutions should provide free data and virtual private network (VPN) access in order to minimize the disruption to office work during major infectious diseases.

## 4. Conclusions

In conclusion, the strength-opportunity (SO) strategy includes the continuous reinvention of the health emergency system, emphasis on people-oriented policies, and upgrading of the health emergency information system. The weakness-opportunity (WO) strategy includes the formulation of a health emergency response system for major infectious diseases during the holidays, the establishment of health emergency specialty and emergency management departments at universities, and the development of a public health emergency management system and code of conduct. The strength-threat (ST) strategy includes authoritative departments releasing real information through a centralized system with a timely review system, the timely adjustment of economic structure, strengthening of international and domestic regional links, enhancement of scientific research relating to major infectious diseases, and facilitating full development of the functions of the medical and health system. The weakness-threat (WT) strategy includes strengthening the public’s psychological interventions in response to public health emergencies, formulating plans for returning to work in different industries, increasing support for health emergency education, and promoting mobile work during major infectious disease epidemics. There is still much work to be done for the prevention and control of COVID-19.

In short, based on the SWOT analysis of the COVID-19, we have integrated the relevant factors that are currently scattered, and have a more systematic and intuitive strategy for the prevention and control of COVID-19 in China. We combed SO, WO, ST, and WT strategies. We performed an in-depth analysis of the policy’s highest-priority areas that we identified, and the highest-priority policies are as follows: continuing to reshape the health emergency system; establishing health emergency departments in universities, and health emergency management departments in all institutions; adjusting the economic structure and strengthening international and domestic linkages; and strengthening public intervention in responding to public health emergencies. From the perspective of continuously responding to the global public crisis, the continuous reconstruction of the health emergency system should attract sufficient attention from countries around the world.

## Figures and Tables

**Figure 1 ijerph-17-02235-f001:**
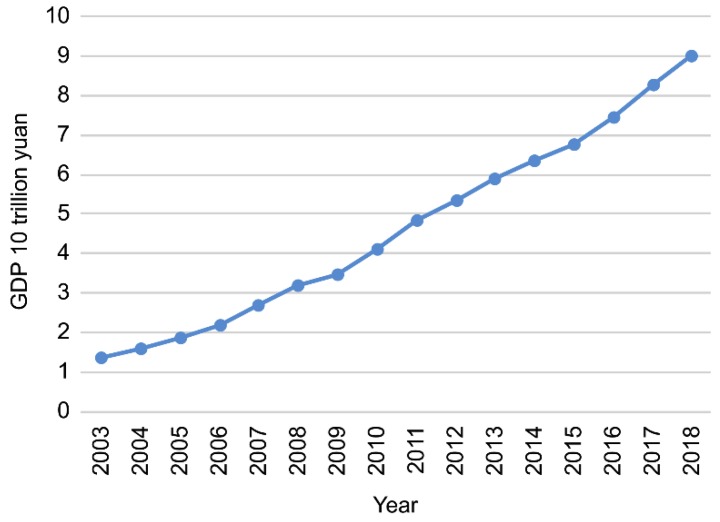
2003–2018 gross domestic product (GDP) of China.

**Figure 2 ijerph-17-02235-f002:**
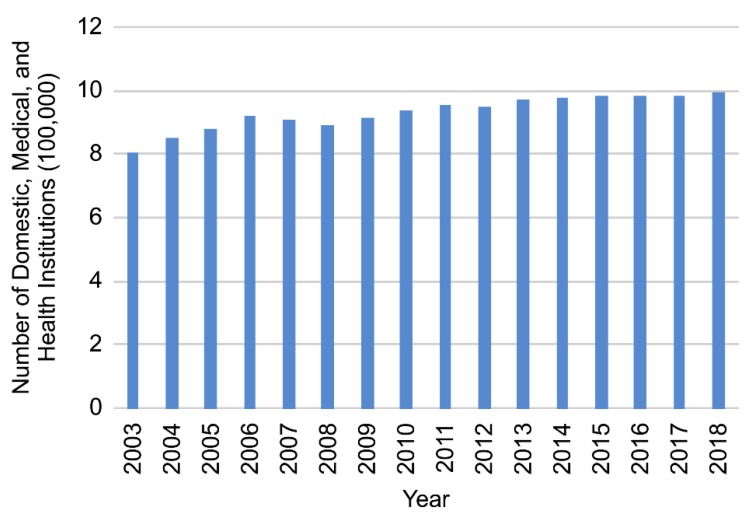
Number of domestic medical and health institutions, 2003–2018.

**Figure 3 ijerph-17-02235-f003:**
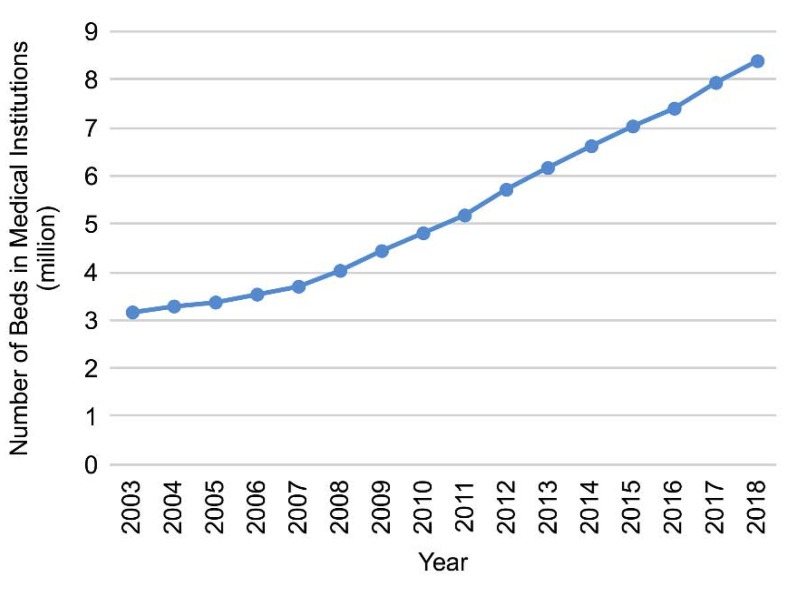
Number of beds in medical institutions, 2003–2018.

**Figure 4 ijerph-17-02235-f004:**
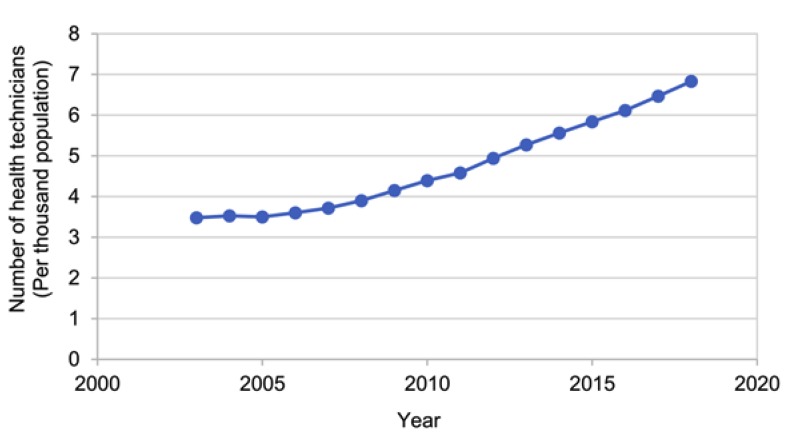
Number of health technicians, 2003–2018 (per thousand population).

**Figure 5 ijerph-17-02235-f005:**
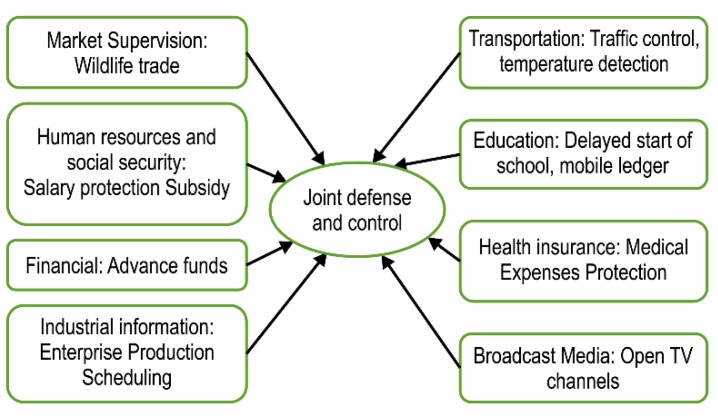
**Coronavirus** (COVID-19) epidemic joint prevention and control in various departments.

**Table 1 ijerph-17-02235-t001:** Development of China’s health emergency system after the SARS epidemic.

Plan	System	Mechanism	Legal
National Emergency Public Emergency Plan, January 26, 2005 [10]; National public emergency medical emergency plan, February 26, 2006 [11]; National Public Health Emergency Response Plan, February 28, 2006 [12]	In 2005, a health emergency management system based on “unified leadership, comprehensive coordination, classified management, hierarchical responsibility, and territorial management” was established [13]	In 2006, established a rapid response mechanism for disease prevention and control, and a national and provincial or regional emergency response mechanism for public health emergencies; In 2016, a joint prevention and control mechanism was established, and health emergency coordination mechanisms at different levels in adjacent areas were established [14]	Emergency Regulations on Public Health Emergencies (Effective May 9, 2003) [15] Law of the People’s Republic of China on Prevention and Control of Infectious Diseases (Implemented on December 1, 2004, Revised June 29, 2013) [16]; Law of the People’s Republic of China on Emergency Response (Implemented on November 1, 2007) [17]

**Table 2 ijerph-17-02235-t002:** Outbreak scope.

Content	Epidemic	
	SARS	Coronavirus (COVID-19)
Epidemic area	Foshan, Guangdong Province	Wuhan, Hubei
Outbreak time (number of cases)	November 16, 2002 (first case)	December 31, 2019 (27 cases)
Spread time across provinces/regions across the country (up to time and number of regions)	3 months (as of February 17, 2003, more than 20 provinces or territories)	1 month (as of January 31, 2020, 34 provinces or territories)
Spread time across the world’s countries (up to date and number of countries)	9 months (as of August 7, 2003, SARS cases reported in 30 countries)	Not yet determined (as of 11 March, 2020, 106 countries report COVID-19 cases)

**Table 3 ijerph-17-02235-t003:** SWOT analysis of key factors for COVID-19 epidemic prevention and control strategy in China.

Factor	Content						
Strengths	1. The medical and health system is gradually improving	2. Comprehensive advancement of the health emergency system	3. COVID-19 epidemic Quick and effective cooperation of departmental joint prevention and control				
Weaknesses	1. The COVID-19 epidemic spread to many regions in a short time period	2. COVID-19 epidemic is approaching the Spring Festival, and epidemic prevention and control measures are more complicated	3. China is a vast country with a huge population	4. Lack of relief supplies and manpower	5. Health emergency is not established as a discipline	6. The public are flustered and lack awareness	7. Rumors spreading misinformation
Opportunities	1. COVID-19 new exploration	2. Further improvement and inspection of the health emergency system	3. Opportunities for Education for Infectious Diseases				
Threats	1. COVID-19 unknown	2. Impact on the daily life, work, and psychology of the public	3. Impact on the national economy				

Note: strengths (S), weaknesses (W), opportunities (O), and threats (T).

**Table 4 ijerph-17-02235-t004:** COVID-19 strategic opportunity window analysis model.

			S	W
		Internal environment	S1: The medical and health system is gradually improving S2: Comprehensive advancement of the health emergency system S3: COVID-19 epidemicQuick and effective cooperation of departmental joint prevention and control	W1: The COVID-19 epidemic affects many regions in a short time W2: COVID-19 epidemic is approaching the Spring Festival, and epidemic prevention and control measures are more complicated W3: China is a vast country with a huge population W4: Lack of relief supplies and manpower W5: Construction of health emergency disciplines lags behind W6: The public are flustered and lack awareness W7: Lots of information rumors
	Strategic analysis			
External environment				
O			SO	WO
O1: COVID-19 new exploration O2: Further improvement and inspection of the health emergency system O3: Opportunities for Education for Infectious Diseases			SO1: Health emergency system continues to reshape SO2: People-oriented, value policy orientation SO3: Integration and upgrade of the health emergency information system	WO1: Formulate a health emergency response system for major infectious diseases on holidays WO2: Health emergency departments are added to universities, and health emergency management departments are added to all institutions WO3 Construction of health emergency culture and code of conduct system
T			ST	WT
T1: COVID-19 unknown T2: Impact on the daily life, work, and psychology of the public T3: Impact on the national economy			ST1: Authoritative department timely releases real information centralization system and review system ST2: Promptly adjust economic structure and strengthen international and domestic linkages ST3: Strengthen scientific research and transformation of major infectious diseases ST4: Fully develop the functions of the medical and health system	WT1: Strengthen public intervention in responding to public health emergencies WT2: Formulate return to work plans for different industries WT3: Increase support for health emergency education WT4: Universal mobile office during major infectious diseases

SO: strength-opportunity; WO: weakness-opportunity; ST: strength-threat; WT: weakness-threat.

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
