# Peer review of "Strengths, Weaknesses, Opportunities and Threats (SWOT) Analysis of China’s Prevention and Control Strategy for the COVID-19 Epidemic"

_ijerph, 2020, doi:10.3390/ijerph17072235_

Round 1

Reviewer 1 Report

Major:

In principal this is a very intriguing manuscript as a SWOT analysis is an excellent choice of tool for analysis of an existing system such that strategic opportunities can be identified and then prioritized. In its execution this SWOT analysis gets bogged down however and does not result in an actionable prioritized plan wherein in scientific support for any prioritization is provided. A deeper analysis of the top priorities for a strategy resulting from the SWOT analysis exercise is missing. For every element in the 4 SWOT boxes on the Table 3 there is a matching SO, WO, ST, WT, etc. in an almost mechanical fashion. Perhaps it is simply too early for this type of window analysis of the SWOT considering this is an ongoing epidemic. Identification of key lessons learned from the SARS outbreak that were implemented successfully as evidenced by the COVID-19 oubreak responses to date is of value and likely something that can be cogently written about. Identification of new weaknesses, unsuspected due to the specific characteristics of this current outbreak, and thereby what opportunities exist for future improvement is also of value. However, the all encompassing approach attempted here appears premature and too generic in its implementation to be particularly useful. There are some astute observations buried in this manuscript that are certainly worthwhile but you have to dig to extract them.

Throughout due to a need for further language editing this manuscript is difficult to interpret. There is jargon that is ill-defined and repetition in other areas that is unnecessary.

References for the analysis approach taken are lacking. I also had difficulty accessing some of the references, e.g. Science Press's website was non-responsive. There is a lack of references to the primary public health literature focused on emergency health system infrastructure, outbreak management, etc. 

Minor (examples of corrections required throughout):

p. 1 Line 21 page 1: What is a "real information centralization system?" 

p. 1 Line 30 - provide the date for the WHO announcement. 

p. 4 Line 83 - "rapid" not needed

p. 4 Line 85 Please provide further explanation as to what the "one plan, three systems" strategy is.

p. 4 Line 89 Do you mean "infrastructure" where "construction system" is used?

p. 5 "One plan, three systems" landmark construction effect - this doesn't header doesn't make any sense to me; more explanation is needed for this jargon.

Table 1 would probably be more readable turned on its axis, i.e. headers of SARS experience vs. post-SARS across the top and rows for organization, plan, etc. It is a difficult presenting this as a table overall though as what is apparent is that there was nothing in place in any category at the time of the SARS outbreak and then an effort was made post-SARS to address the issues in the system. It may be best to simply summarize the absence of these things and the lessons learned and then describe what has been put in place rather than trying to present it as such an imbalanced table. 

p. 5 Line 93 China "has" a centralized...

p. 6 Fig 5 - what is "other department"

p. 7 Line 142 - "coincided"

p. 7 line 150 - please define "floating population"

p. 8 line 177 instead of health emergency disciplines, "emergency medicine" may be a better term. And again the use of "construction" is odd

p. 8 line 204-205 last sentence in paragraph suggested replacement "None of these rumors have any solid foundation in science or medicine."

p. 9 208-209 How can "infectivity" be a "walking source of infection?"

p. 9 211 "The degree of harm posed by 2019-nCoV to different populations is constantly changing." - This sentence doesn't make any sense to me. What do you mean - the virus is not necessarily changing. 

The incubation period is not well defined? There seem to be a lot of statements made here with no references. And also this paragraph gets into highly subjective territory e.g. "Chinese scientific researchers are unremitting... Please revise this paragraph to simply present facts and unknowns about the 2019-nCoV virus. The last sentence is ok.

Since this is an outbreak in progress, please update numbers and other statistics before resubmission.

p 9 line 222 "emergency health system" 

p 10 line 257 "isolated at home"

p. 10 line 261 "consumption?" A term for the industries listed here is "hospitality"

p. 10 line 263 

Author Response

Manuscript ID: ijerph-734022 (SWOT-CLPV analysis of China's prevention and control strategy for the COVID-19 epidemic)

Response to Reviewer #1

The revised portions of the manuscript are indicated in red font. 

Major:

In principal this is a very intriguing manuscript as a SWOT analysis is an excellent choice of tool for analysis of an existing system such that strategic opportunities can be identified and then prioritized. In its execution this SWOT analysis gets bogged down however and does not result in an actionable prioritized plan wherein in scientific support for any prioritization is provided. A deeper analysis of the top priorities for a strategy resulting from the SWOT analysis exercise is missing. For every element in the 4 SWOT boxes on the Table 3 there is a matching SO, WO, ST, WT, etc. in an almost mechanical fashion. Perhaps it is simply too early for this type of window analysis of the SWOT considering this is an ongoing epidemic. Identification of key lessons learned from the SARS outbreak that were implemented successfully as evidenced by the COVID-19 outbreak responses to date is of value and likely something that can be cogently written about. Identification of new weaknesses, unsuspected due to the specific characteristics of this current outbreak, and thereby what opportunities exist for future improvement is also of value. However, the all-encompassing approach attempted here appears premature and too generic in its implementation to be particularly useful. There are some astute observations buried in this manuscript that are certainly worthwhile but you have to dig to extract them.

Response: Thank you very much for your suggestion. In consideration of your suggestions, we have revised the article very carefully. According to the above SWOT analysis results, we conducted a basic analysis of the strategy, identified important issues and strategies, and then built a strategic opportunity window analysis model. We integrated the current prevention and control strategies of the COVID-19 epidemic into the overall functioning of society in China. We have provided a more systematic and intuitive understanding, that is with, SO, WO, ST, WT strategies. We conducted a more in-depth analysis and data mining and extraction resulted in identification of actionable priority plans to scientifically support the priorities. The highest priority strategies are reflected in SO1, WO2, ST2, and WT1. Therefore, SO1, WO2, ST2, and WT1 can be explained in more depth. The highest-priority policies are as follows: 1) continuing to reshape the health emergency system; 2) establishing health emergency departments in universities, and health emergency management departments in all institutions; 3) adjusting the economic structure and strengthening international and domestic linkages; and 4) strengthening public intervention in responding to public health emergencies.These additional analyses are reported towards the end of the article, in text in red font. In the process of prevention and control of the COVID-19 epidemic, we can systematically adhere to the prevention and control strategy and remain focused. This is a core tenet of health emergency management decision-making. 

Throughout due to a need for further language editing this manuscript is difficult to interpret. There is jargon that is ill-defined and repetition in other areas that is unnecessary.

Response: We have had the manuscript re-edited to improve the language.  

References for the analysis approach taken are lacking. I also had difficulty accessing some of the references, e.g. Science Press's website was non-responsive. There is a lack of references to the primary public health literature focused on emergency health system infrastructure, outbreak management, etc. 

Response: We have updated the references section and have added some references and have checked that the online references are accessible, including several references from a recent special issue of The Lancet.  

Minor (examples of corrections required throughout):

  1. 1 Line 21 page 1: What is a "real information centralization system?" 

Response: Thank you very much for your suggestion. It means an authoritative information system. We have revised the Abstract in Line 16-21 page 1.

  1. 1 Line 30 - provide the date for the WHO announcement. 

Response: We have added the date in Line 28 page 1. 

  1. 4 Line 83 - "rapid" not needed

Response: We have deleted the word “rapid” in Line 86 page 4. 

  1. 4 Line 85 Please provide further explanation as to what the "one plan, three systems" strategy is.

Response: We have added that “one plan” is an emergency plan, and that “three systems” refers to establishing and improving emergency response systems and mechanisms and revising the legal system in Line 91-93 page 4. 

  1. 4 Line 89 Do you mean "infrastructure" where "construction system" is used?

Response: This sentence is redundant, so we have deleted it in Line 95 page 4. 

  1. 5 "One plan, three systems" landmark construction effect - this doesn't header doesn't make any sense to me; more explanation is needed for this jargon.

Response: We have changed the orientation of the tables in portrait as it would maintain the consistency of the article as per the template in Table1 of Line 96 page 4. 

Table 1 would probably be more readable turned on its axis, i.e., headers of SARS experience vs. post-SARS across the top and rows for organization, plan, etc. It is a difficult presenting this as a table overall though as what is apparent is that there was nothing in place in any category at the time of the SARS outbreak and then an effort was made post-SARS to address the issues in the system. It may be best to simply summarize the absence of these things and the lessons learned and then describe what has been put in place rather than trying to present it as such an imbalanced table. 

Response: Thank you very much for your suggestion. We have changed Table 1 and added descriptive language in Line 87-89 and Table1 of Line 96 page 4. 

  1. 5 Line 93 China "has" a centralized...

Response: We have made this change in Line 99 page 5. 

  1. 6 Fig 5 - what is "other department"

Response: “Other department” means other departments that are not listed by name. These include the Publicity Department, and the Foreign Affairs Department. We have removed this “Other department” from the figure to avoid confusion in Line 131-133 page 5. 

  1. 7 Line 142 - "coincided"

Response: Thank you for the comment. We have made the change in Line 147 page 6.

  1. 7 line 150 - please define "floating population"

Response: We have defined "floating population" as including temporary residents, registered tourists, and transient populations in Line 154-155 page 6. 

  1. 8 line 177 instead of health emergency disciplines, "emergency medicine" may be a better term. And again the use of "construction" is odd

Response: We have changed this to “Health emergency discipline is underdeveloped.” in Line 178 page 7 

  1. 8 line 204-205 last sentence in paragraph suggested replacement "None of these rumors have any solid foundation in science or medicine."

Response: Thank you very much for your suggestion. We have rephrased the last sentence according to your suggestion in Line 204 page 7.

  1. 9 208-209 How can "infectivity" be a "walking source of infection?"

Response: This was meant in a metaphorical sense. We have deleted this sentence in Line 207 page 7. 

  1. 9 211 "The degree of harm posed by 2019-nCoV to different populations is constantly changing." - This sentence doesn't make any sense to me. What do you mean - the virus is not necessarily changing. 

The incubation period is not well defined? There seem to be a lot of statements made here with no references. And also this paragraph gets into highly subjective territory e.g. "Chinese scientific researchers are unremitting... Please revise this paragraph to simply present facts and unknowns about the 2019-nCoV virus. The last sentence is ok.

Response: Thank you very much for your suggestion. We have deleted the sentence “The degree of harm posed by 2019-nCoV to different populations is constantly changing.” and have replaced it with the following sentence: “Severe acute respiratory syndrome coronavirus 2 (SARS-CoV-2), the virus that causes COVID-19, belongs to a new coronavirus independent evolutionary branch. It belongs to the beta coronavirus genus with SARS coronavirus and MERS coronavirus, but has a long genetic distance from the other two highly pathogenic coronaviruses, and its nucleic acid homology is <80%. The virus that is closest to SARS-CoV-2 is a coronavirus that was isolated from the Yunnan chrysanthemum bat, which has a nucleic acid homology of 96%. Therefore, the chrysanthemum bat may be the original source of the virus, but the direct source of the virus has not been found.” in Line 208-215 page 7 and page 8.

Since this is an outbreak in progress, please update numbers and other statistics before resubmission.

Response: We have updated the numbers and other statistics in the revised manuscript in Line 43-44 page 2, Table 2 Line 142 page 6 and Line 174-175 page 7.

p 9 line 222 "emergency health system" 

Response: We have made this change in Line 223 page 8.

p 10 line 257 "isolated at home"

Response: We have made this change in Line 255 page 9.

  1. 10 line 261 "consumption?" A term for the industries listed here is "hospitality"

Response: We have changed this to "service industry.” in Line 259 page 9.

  1. 10 line 263 

Response: We have rephrased this in accordance with your suggestion in Line 261 page 9.

Reviewer 2 Report

The paper addresses an important topic and is well organized and comprehensively written.

The structure and approaches are well documented and the topic has been studied from various angles.

Minor points:

  • The term SWOT analysis is explained in the abstract, but not in the introduction, please add few sentences about the method in the main text
  • Figures 1-4 should be corrected or normalized to the increase in population
  • Please explain the “one plan, three systems” approach in a few sentences in the text
  • Better formatting of tables would be helpful for the reader (e.g. smaller font size)
  • Please provide some information on accuracy of testing methods and the accuracy of numbers of diseased patients

Author Response

Manuscript ID: ijerph-734022 (SWOT-CLPV analysis of China's prevention and control strategy for the COVID-19 epidemic)

Response to Reviewer #2 Comments and Suggestions for Authors

The paper addresses an important topic and is well organized and comprehensively written.

The structure and approaches are well documented, and the topic has been studied from various angles.

Response: Thank you very much for reviewing our manuscript and for your positive feedback and constructive suggestions. We greatly appreciate the time and effort that you devoted to the review and your contribution to improving our manuscript for publication. We have considered all your comments and suggestions and have revised the manuscript accordingly. The revised portions of the manuscript are shown in red font. Our itemized response to your comments is given below.

Minor points:

  • The term SWOT analysis is explained in the abstract, but not in the introduction, please add few sentences about the method in the main text

Response: We have explained this in the main text in Line 62-65 page 2.

  • Figures 1-4 should be corrected or normalized to the increase in population

Response: We have modified the figures1-4 accordingly in Line 74-84 page 2 to 4.

  • Please explain the “one plan, three systems” approach in a few sentences in the text

Response: We have added “one plan” is an emergency plan; “three systems” refers to establishing and improving emergency response systems, mechanisms, and legal systems in Line 91-93 page 4.

  • Better formatting of tables would be helpful for the reader (e.g. smaller font size)

Response: We have the modified tables in Table 1 Line 96 page 4, Table 2 Line 142 page 6, Table 3 Line 266-268 page 9, and Table 4 Line 551 page 16-17.

  • Please provide some information on accuracy of testing methods and the accuracy of numbers of diseased patients

Response: We have updated data and have added some references in Line 55-65 page 2, Line 208-215 page 7 and page 8.